# Use of Enzymatically Activated Carbon Monoxide Donors for Sensitizing Drug-Resistant Tumor Cells

**DOI:** 10.3390/ijms241411258

**Published:** 2023-07-09

**Authors:** Federica Sodano, Barbara Rolando, Loretta Lazzarato, Costanzo Costamagna, Mariacristina Failla, Chiara Riganti, Konstantin Chegaev

**Affiliations:** 1Department of Pharmacy, “Federico II” University of Naples, 80131 Naples, Italy; federica.sodano@unina.it; 2Department of Drug Science and Technology, University of Torino, 10125 Torino, Italy; barbara.rolando@unito.it (B.R.); loretta.lazzarato@unito.it (L.L.); mariacristina.failla@unito.it (M.F.); 3Department of Oncology, University of Torino, 10125 Torino, Italy; costanzo.costamagna@unito.it (C.C.); chiara.righanti@unito.it (C.R.)

**Keywords:** carbon monoxide, carbon monoxide releasing molecules (CORMs), multidrug resistance, antitumor activity, mitochondrial damage

## Abstract

The application of gaseous signaling molecules like NO, H_2_S or CO to overcome the multidrug resistance in cancer treatment has proven to be a viable therapeutic strategy. The development of CO-releasing molecules (CORMs) in a controlled manner and in targeted tissues remains a challenge in medicinal chemistry. In this paper, we describe the design, synthesis and chemical and enzymatic stability of a novel non-metal CORM (**1**) able to release intracellularly CO and, simultaneously, facilitate fluorescent degradation of products under the action of esterase. The toxicity of **1** against different human cancer cell lines and their drug-resistant counterparts, as well as the putative mechanism of toxicity were investigated. The drug-resistant cancer cell lines efficiently absorbed **1** and **1** was able to restore their sensitivity vs. chemotherapeutic drugs by causing a CO-dependent mitochondrial oxidative stress that culminated in mitochondrial-dependent apoptosis. These results demonstrate the importance of CORMs in cases where conventional chemotherapy fails and thus open the horizons towards new combinatorial strategies to overcome multidrug resistance.

## 1. Introduction

Despite enormous efforts undertaken in recent decades and better understanding of pathophysiology, cancer remains the second leading cause of death worldwide and is the first in developed countries. Several types of treatments are available for tumors, such as surgery, chemotherapy, radiation therapy, and immunotherapy, which can be used alone or in combination [1]. Among these, chemotherapy is still the method of first choice for cancer treatment; however, the emergence of resistant drugs could seriously undermine its effectiveness [2]. Currently, cancer cells can develop cross-resistance to several drugs with a different mechanism of action, giving rise to multidrug resistance (MDR). Therefore, MDR represents a serious obstacle to the use of chemotherapeutic agents. Further efforts are needed to overcome the development of MDR and increase tumor susceptibility to chemotherapeutic drugs.

Previously, we have found that some derivatives releasing gaseous signaling molecules, such as nitrogen monoxide (NO) and hydrogen sulfide (H_2_S), could help overcome the resistance of cancer cell to doxorubicin via several mechanisms. Specifically, NO was able to nitrate the tyrosine residues of multidrug resistance-related (MRP)-3, MRP-5 and P-glycoprotein (Pg-p) proteins located on the cell membrane, to prevent drug efflux of NO-conjugated doxorubicin and gemcitabine, increasing the intracellular accumulation and the cytotoxic potential of the drugs in terms of DNA damage, oxidative and nitrosative mitochondrial stress [3,4,5]. H_2_S was able to interfere with protein folding in the endoplasmic reticulum through their sulfhydration, promoting protein ubiquitination and apoptosis-related endoplasmic reticulum stress [6,7]. Our in vivo studies have also demonstrated that such a strategy could be of great interest in treating cancer and overcoming MDR [8,9]. As a continuation of our research, we decided to investigate another gaseous signaling molecule, carbon monoxide (CO), and study its activity in drug-resistant tumor cell lines of different origin.

CO was first identified in the exhaled air of a healthy human in 1949, but it took a long time to classify CO as an endogenous gas transmitter. Endogenous CO formation is due to the activity of a family of enzymes called heme oxygenases (Hos). So far, three isoforms of HOs have been identified: HO-1, HO-2 and HO-3 [10]. The first two isoforms transform heme to biliverdin, ferrous iron and CO, while the function of the third isoform is still under investigation. The physiological action of CO affects many systems like nervous, immune, cardiovascular, gastrointestinal, reproductive, and respiratory systems [11]. Recently, it has also attracted great attention as a therapeutic agent in cytoprotection [11,12,13,14]. Similar to NO and H_2_S, CO also has both pro-tumoral and anti-tumoral effects, depending on the concentration in tumor microenvironment. Low levels of CO, not toxic at systemic levels, inhibit metastasis and block the mitochondrial electron transport chain (ETC) [15], inducing mitochondrial stress and increasing the intracellular levels of reactive oxygen species (ROS) [16]. It has been seen that treatment of tumor cells with CO accelerates mitochondrial respiration, thus leading to metabolic depletion and mitochondrial collapse [17]. Based on this knowledge, controlled administration of CO to tumor cells can destroy mitochondria and induce depletion of energetic metabolism, dramatically decreasing ATP synthesis in tumor cells and consequently the efflux of drugs via ABC transporters that is fueled by mitochondrial ATP [18]. Although CO can affect a few physiological and pathophysiological processes, the use of gaseous CO is dramatically complicated by its lethality due to inhibition of oxygen transport, difficulties in its safe administration and lack of selectivity for any tissue. In this context, the controlled delivery of CO to target tissue remains a challenge in medicinal chemistry. Several CO-releasing molecules (CORMs) have been proposed as alternatives to direct CO administration [19]. Among them, carbonyl complexes of transition metals (such as Fe, Mn, and Ru) have been extensively studied for their commercial accessibility and stability. However, the presence of transitional metals could increase the toxicity of CORMs and generally, the complexity of pharmaceutical development [20,21]. In addition, metal carbonyls could undergo ligand exchange processes resulting in the release of background CO [21]. Metal-free CORMs could be a valuable alternative to metal carbonyls without these drawbacks.

Metal-free CORMs are small organic molecules that can release CO when triggered by enzymes, light or magnetic heating. Moreover, they could be easily modified to modulate their physicochemical properties and conjugated with chemotherapeutic drugs. For the design of our CORM (compound **1**), we used the strategy proposed by Wang et al. [22], which consisted of the synthesis of a cyclopentadienone conjugated to an inactive dienophile. Under the action of enzymes, cleavage of the internal lactone occurred, which activated the dienophile promoting the intramolecular Diels–Alder reaction. The resulting norbornadienone intermediate **2** eliminates CO and produces a fluorescent compound **3** that allowed the release and localization of CO to be followed (Figure 1).

In this study, the chemical and enzymatic stability of **1** to hydrolysis were evaluated at physiological pH and in human serum, as well as its toxicity against different human cancer cell lines and their drug-resistant counterparts. Finally, the putative mechanism of toxicity was investigated.

## 2. Results

### 2.1. Chemistry

The synthesis of the desired CORM was performed as follows (Figure 2). Briefly, 1,5-di *tert*-butyldimethylsilyl-protected pent-2-in-1,4,5-triol **4** was linked to ethyl 2-hydroxy-3-(2-*tert*-butyldiphenylsilyloxy)ethyl benzoate **5** under Mitsunobu conditions. After hydrolysis of *tert*-butyldimethylsilyl-protected groups and ester moieties (**7**), 7-membered lactone ring was closed using acetic anhydride at reflux (**8**). Further removal of *tert*-butyldiphenylsilyl group was carried out under the action of HF⋅Py complex, and the obtained phenylethanol **9** was oxidized with Jones reagent to yield phenylacetic acid **10**. The coupling reaction of **10** with Meldrum’s acid under the action of EDCI and DMAP and further treatment of **11** with MeOH in toluene under reflux yielded β-ketoester **12**. Finally, reaction of **12** with acenaphtenechinone in acetic anhydride catalyzed by sulfuric acid yielded the desired cyclopentadienone structure of CORM (**1**).

### 2.2. Stability Studies

The hydrolytic susceptibility of our compound **1** was studied under different conditions. We first tested the chemical stability of **1** at physiological pH, using phosphate-buffered solution (PBS): specifically, compound **1** was incubated at pH 7.4 in PBS at 37 °C and quantified using HPLC analysis at different time over 24 h. Under these conditions, as expected, CORM was completely stable; in fact, the degradation rate was considered negligible. Stability was then evaluated in human serum: compound **1** was incubated at 37 °C in esterase-rich medium and quantified under the same chromatographic conditions at different time over 240 min. In this esterase-rich medium, CORM (**1**) turned into a series of fluorescent compounds (Appendix A), degrading with pseudo-first-order kinetics, and displaying a half-life of 110 min (Figure 1).

### 2.3. Biological Studies

#### 2.3.1. CORM Is Effectively Taken Up by Drug-Resistant Cancer Cells Lines and Overcomes Resistance to Multiple Chemotherapeutic Drugs

First, we tested the ability of **1** to be taken up by human cancer cell lines of different histological origin (non-small cell lung cancer, NSCLC; triple negative breast cancer, TNBC; and pancreatic adenocarcinoma, PDAC), known for their intrinsic resistance to chemotherapy caused by the high levels of different ABC transporters, namely ABCB1, ABCC1, and ABCC5 present on the cell surface of each cell line, measured via flow cytometry (Table 1).

In all the six cell lines analyzed, **1** was taken up in a time-dependent way (Figure 2), albeit with cell type-dependent kinetics. Indeed, we detected a progressively increasing intracellular fluoresce. Although indirect, this is an index of the progressive release of CO from **1**, since the fluorescence measured was caused by the products derived from the hydrolysis of compound **1** and consequent CO release.

Next, we evaluated the cytotoxicity of compound **1** in tumor cell lines and corresponding non-transformed epithelial cells, namely epithelial bronchial BEAS-2B, epithelial mammary MCF10A and epithelial pancreatic HPDE. For both transformed and non-transformed cells, we observed a dose-dependent decrease in cell viability, which became significant at 1 µM (for NSCLC) or 0.1 µM (for TNBC and PDAC). Interestingly, toxicity in non-transformed cells, although present, was significantly lower than in the corresponding cancer cells (Figure 3), suggesting a selective killing of tumor cells exerted by our CORM.

Based on these results, for subsequent co-administration with chemotherapeutic drugs, we chose the concentration of 1 µM, corresponding to the approximate IC_25_ value in each cancer cell line, resulting in mild toxicity. When compound **1** was incubated with increasing concentrations of cisplatin (Pt) in NSCLC, doxorubicin (Dox) and docetaxel (Dtx) in TNBC, and gemcitabine (Gem) in PDAC, i.e., the first line-treatments for these tumor types notwithstanding their intrinsic resistance to these chemotherapeutic drugs (Table 1), we observed that chemotherapy alone dose-dependently decreased cell viability, but even 10 µM of each drug did not reduce cell viability more than 50%. Conversely, co-administration of compound **1** produced a more pronounced decrease in cell viability, significantly higher than the decrease exerted by the chemotherapeutic drug alone, in nanomolar-low micromolar range of each chemotherapeutic drug (Figure 4A–D). Notably, our CORM reduced the catalytic efficiency of the ABC transporters mostly expressed in each resistant cell lines, namely ABCC1, which transports cisplatin in NSCLC cells; ABCB1, the effluxer of doxorubicin and docetaxel in TNBC and ABCC5, the gemcitabine transporter in PDAC cells (Figure 4E). This effect may justify the increased cell death achieved by co-incubation of chemotherapeutic agents with CORM, which may reduce the efflux of the chemotherapeutic agents from the resistant cells.

#### 2.3.2. CORM Alters Mitochondrial Energy Metabolism and Enhances Chemotherapy-Dependent Mitochondrial Apoptosis in Resistance Cells

Since CO could exert mitochondrial damage [12], we investigated whether the killing of resistant cancer cells could be due to alterations in mitochondria-related parameters. In all cell lines, CORM alone reduced electron flux through the ETC (Figure 5A), resulting in reduced ATP production (Figure 5B). This metabolic crash was sufficient to induce mitochondrial depolarization (Figure 5C) and a slight, but significant, increase in TNBC and PDAC mitochondrial permeability transition pore (mPTP) opening (Figure 5D). In line with this scenario, our CORM increased caspase-9 and caspase-3 activation (Figure 5E,F) in TNBC and PDAC cells, but not in NSCLC cells, where mitochondrial depolarization was not sufficient to open the mPTP. Although cisplatin [27], doxorubicin [28], docetaxel [29] and gemcitabine [30] have been reported to alter energetic mitochondrial metabolism and integrity, none of these agents used at 1 µM, i.e., a concentration that induced an average reduction in cell viability of 25% (Figure 4) elicited significant mitochondrial damage (Figure 5A–F; Appendix A), which is consistent with the drug-resistant phenotype of these cell lines. Notably, the combination of the mild mitochondrial-damaging agent CORM and chemotherapeutic drugs, which were ineffective on mitochondria, dramatically impaired all mitochondrial parameters, decreasing ETC and ATP production, increasing depolarization and mPTP opening, and triggering the caspase-9/caspase-3 axis (Figure 5A–F, Appendix A). Most importantly, the changes induced by the combination of CORM and chemotherapeutic drugs are significant not only compared with untreated cells but also with respect to chemotherapy alone.

#### 2.3.3. CORM-Induced Chemosensitization Is Caused by Increased Oxidative Stress and CO Release

Since CO-mediated inhibition of mitochondrial oxidative metabolism can increase the amount of mitochondrial (mt) ROS, a typical phenotype observed with an impaired ETC in cancer cells [23], we measured ROS in mitochondrial extracts and thiobarbituric reactive substances (TBARS), i.e., lipoperoxidation products that indicate the presence of oxidative stress. Compound **1** increased both mtROS and mtTBARS, whereas cisplatin, doxorubicin, gemcitabine, and docetaxel are devoid of effects, although they have been reported to induce oxidative stress in sensitive cells [31,32,33,34] (Figure 6A,B, Appendix A). A similar increase in ROS and lipoperoxidation was observed in whole cell lysates (Figure 6C,D, Appendix A). The combination of CORM and each chemotherapeutic agent tremendously boost these two oxidative stress parameters, which were instead completely abrogated by red blood cells, used as a CO scavenger, and by mitoquinol, a mitochondrial-derived ROS scavenger [23]. The increase in mitochondrial and total ROS and TBARS was significantly higher in cells treated with combination compared to cells treated with CORM alone, indicating that the chemotherapeutic drug rescues its ability to induce an oxidative stress, enhancing the stress elicited by CORM.

Finally, to prove CO release and subsequent mitochondrial ROS production, we re-assessed the cell viability of compound **1**, associated with cisplatin, doxorubicin, docetaxel or gemcitabine, in cells co-treated with RBC or mitoquinol. The two scavengers abrogated the cytotoxic effects of the combinations CORM plus chemotherapeutic drugs (Figure 7A–D), suggesting that the cell death elicited by these combinations in resistant cells is mediated by increased CO and mtROS.

## 3. Discussion

The use of a gaseous transmitter CO in the cancer treatment have garnered great attention in recent years. To overcome the difficulties associated with the use of CO in a gaseous form, several molecules capable of releasing CO in the physiological environment have been proposed, among them carbonyl complexes of transition metals are the most used. These complexes are able to release CO under certain triggers, however, the toxicity associated with transition metals complicates their potential clinical use. To avoid this problem, metal-free organic molecules, capable of releasing CO under certain conditions have been developed.

Compound **1** was designed to be activated by esterase enzymes catalyzed lactone ring opening followed by the release of carbon monoxide molecule upon intermolecular transformation, while formation of fluorescent products could facilitate the localization and kinetics measurements of the CO release. The presence of a carboxyl group in the molecule allows the synthesis of multitarget compounds in the future where the CORM moiety will be chemically conjugated to a chemotherapeutic drug.

Synthesized CORM **1** was stable under physiological conditions but hydrolyzes slowly with the consequent release of CO in human serum. The half-life of **1** in human serum was found to be nearly two hours.

Interestingly, our CORM was taken up in all the cell lines analyzed, irrespective of the pattern of different ABC transporters present, suggesting that it was not a substrate peculiar to these pumps. The different kinetics of entry, accumulation and metabolism are expected when considering the different epithelial origin of the cancer cell lines, which implies different membrane composition and different importers [35] that may favor or hamper CORM uptake, different pattern of enzymes that can metabolize CORM into the fluorescent products of hydrolysis and CO, and/or the differences in the endosomal–lysosomal system that can sequester or release the compound after its uptake [36].

Our experimental viability set indicated that compound **1** was more toxic to cancer cells than for non-transformed cells, indicating the existence of a possible therapeutic window for future in vivo applications. All cell lines tested were resistant to their first-line treatment, i.e., cisplatin for NSCLC, doxorubicin and docetaxel for TNBC, and gemcitabine for PDAC, as evidenced by the low cytotoxicity exerted by the drugs alone, except at micromolar concentrations, which are hardly achievable in patients. This intrinsic resistance, although pleiotropic, is also explained by the presence of specific ABC transporters in each cell line. Indeed, ABCB1 and ABCC1 are the main transporters of cisplatin; ABCB1 is critical in extruding doxorubicin and docetaxel, and ABCB5 transports gemcitabine [37]. On the other hand, the presence of ABC transporters did not affect the efficacy of CORM, which enhanced the cytotoxicity of chemotherapeutic drugs even in cells expressing their major efflux proteins. Our results are of particular relevance because they demonstrate that a CORM reverses resistance to drugs that are commonly used as first-line treatment in their respective tumors but often fail due to intrinsic tumor resistance. We are aware that future validations in preclinical models are needed to verify the in vivo efficacy and safety of these combinations. The present results, however, proves the concept that CO is a novel chemosensitizing agent that acts across a broad spectrum of chemotherapeutic drugs.

Moreover, the different sensitivity between cancer and non-transformed cells may also suggest that CO targets specific metabolic or signaling pathways necessary for the survival of resistant cancer cells more than for the survival of normal cells. We focused on mitochondrial energetic metabolism because we have already demonstrated that phenol derivatives synthesized and conjugated with triphenylphosphonium (TPP) cation, a mitochondria-targeted vector, selectively killed cancer cells compared with non-transformed cells because the former had higher active mitochondrial metabolism and greater dependence on oxydative phosphorylation (OXPHOS) to fulfill their energy demands [38]. Indeed, TPP derivatives reduced mitochondrial mass and OXPHOS, caused a crash in mitochondrial ATP and an increase in mitochondrial ROS, culminating in mitochondria depolarization and caspase-9-mediated apoptosis [38]. Compared with sensitive cells, chemoresistant cells exhibited high mitochondrial energetic OXPHOS-based metabolism [23,39], but at the same time a derangement of OXPHOS metabolism is an Achille’s heel for chemoresistant cells. This is the case with NO-releasing doxorubicins that have mitochondrial tropism and induce mitochondrial nitrosative stress [4,8]. CORM had slightly different effects on different cell lines: in fact, it produced a minor reduction in ETC and ATP in NSCLC cells, which are the most OXPHOS-based cell lines among those analyzed. In line with this small reduction, we did not detect sufficient damage to trigger mitochondrial-dependent apoptosis, as indicated by the absence of caspase-9 activation. Hence, in NSCLC cells, the decreased viability of cells treated with the combination of cisplatin and compound **1** is likely due to mitochondrial-independent mechanisms, i.e., the intracellular accumulation of cisplatin caused by the inhibition of ABCC1 by CORM. By contrast, CORM was more effective in TNBC and PDAC cells that had lower OXPHOS metabolism: in this case, compound **1** produced a significant decrease in ETC and ATP, inducing severe depolarization and triggering mitochondrial damage-dependent apoptosis. Targeting OXPHOS and inducing energetic collapse is an effective strategy to overcome drug resistance, because chemoresistant cells rely on mitochondrial ATP and oxidoreductive cofactors more than chemosensitive cells [39] and mtATP fuel drug efflux transporters [18]. In previous work on chemoresistant colon, breast, lung cancer and osteosarcoma, micromolar dosing of chemotherapeutic agents has been sufficient to induce cell death in sensitive but not in resistant cells [32,39,40]. This was also true with regard to mitochondria metabolism: micromolar doses of doxorubicin impaired ETC and mtATP induced mitochondrial depolarization and cell death by apoptosis in highly doxorubicin-resistant colon and breast cancer cells [32]. For this reason, we worked with 1 μM concentration of each drug but did not alter any mitochondrial parameters. These results are in line with the low induction of cell death and can be attributed to the intrinsic nature of the high chemoresistance of the analyzed cell line, which has one or more ABC transporters effluxing drugs. Notably, the combination of our CORM and chemotherapeutic drugs instead produced strong mitochondrial damage, culminating in caspase-9-dependent apoptosis. This effect may be attributed to the direct inhibition of mitochondrial complex IV by CO [16] and/or increased accumulation of chemotherapeutic drugs within cells, caused by reduced mtATP production. Alternatively, as already observed for NO and H_2_S [5,6,7], CO may also directly reduce the activity of ABC transporters: it has been reported that CO oxidizes the cysteines of integral membrane proteins, triggering a radical cascade at the membrane protein–lipid interface that alters protein functions [41]. Our CO-donor also induced a reduction in the catalytic efficacy of ABCB1, ABCC1 and ABCC5 in cell lines with the highest levels of these transporters, suggesting a lower efficiency in efflux of the respective chemotherapeutic substrates. Such inhibition may increase the intracellular retention of chemotherapeutic drugs to a level where it can exert mitochondrial oxidative stress that amplifies the damage elicited by compound **1**.

An impaired OXPHOS metabolism is typically associated with increased oxidative stress caused by ROS generation within the mitochondria and diffused into the cells [23]. This situation is also elicited by our CO donor. Oxidative stress is enhanced by the presence of a chemotherapeutic drug that, although ineffective when used alone, determines oxidative damage, such as the agents tested in the present work. Lipoperoxidation is a typical marker of oxidative stress, in addition to being an indicator of biochemical damage on mitochondria and cell membrane integrity. Strong lipoperoxidation has been observed both in the mitochondria fraction and in whole cells. This damage can be caused by mtROS produced or directly by CO, as previously reported [10]. Mitochondrial ROS [16] and lipid peroxidation [41] are two conditions suggestive of ferroptosis, a type of cell death independent of caspase activation but triggered by oxidative stress, often catalyzed by iron or promoted by reduced antioxidant defenses or immune signaling from tumor microenvironment [42]. Because we detected either oxidative stress and lipoperoxidation in cells treated with CORM and chemotherapeutic drugs, we cannot exclude that CORM acted by inducing ferroptotic death. Indeed, mesoporous carbon nanoparticles loaded with doxorubicin and iron carbonyl Fe (FeCO) have been shown to induce ferroptosis in breast cancer cells and at the same time sensitize cells to doxorubicin [43], which is consistent with our results.

Since we also observed the activation of the caspase-9/caspase-3 axis, we suggest that CORM acts according to a pleiotropic mechanism. On one hand, it impairs energetic metabolism and polarization of mitochondria, triggering mitochondrial-dependent apoptosis, and it can exert a second mechanism of cell death, based on ferroptosis. The latter can be induced by increased mtROS producing extensive lipoperoxidation or by a direct effect of CO in terms of inducing oxidative stress and lipoperoxidation. Regardless of the mechanism of cell death, both CO release and ROS production mediate cell death, as demonstrated by the restoration of cell viability elicited either by CO scavengers or ROS scavengers, even in the presence of the most lethal combination of CORM plus chemotherapeutic drug.

To the best of our knowledge, this work is the first demonstration that CO donors can rescue the efficacy of several chemotherapeutic drugs in solid cancers that express ABC transporters and are refractory to clinically used chemotherapy, thereby opening new combinatorial strategies to achieve chemosensitization.

## 4. Materials and Methods

### 4.1. Chemistry

#### 4.1.1. Synthesis

All chemicals used for the synthesis were purchased from commercial sources (Merck KGaA, Darmstadt, Germany). All solvents were purified and degassed before use. Chromatographic separation was carried out under pressure on Merck silica gel 60 using flash-column techniques. Reactions were monitored via thin-layer chromatography (TLC) carried out on 0.20 mm silica-gel-coated aluminum plates (60 Merck F254, Merck KGaA, Darmstadt, Germany). Unless specified, all reagents were used without further purifications. Dichloromethane was dried over P_2_O_5_ and freshly distilled under nitrogen prior to use. THF dry was freshly distilled from sodium/benzophenone. ^1^H and ^13^C NMR spectra were recorded at room temperature on a JEOL ECZ-R 600 instrument at 600 and 150 MHz, respectively, and calibrated using solvent residual peak as an internal reference. Chemical shifts (δ) are given in parts per million (ppm) and the coupling constant (*J*) in Hertz (Hz). The following abbreviations were used to designate the multiplicities: s = singlet, d = doublet, dd = doublet of doublet, dt = doublet of trilet, t = triplet, and m = multiplet. ESI spectra were recorded using a Micromass Quattro API micro (Waters Corporation, Milford, MA, USA) mass spectrometer. Data were processed using a MassLynxSystem v. 4.1 SCN 714 (Waters Corporation, Milford, MA, USA). The purity of the final compound (**1**) was determined via RP-HPLC analysis performed using the HP1100 chromatograph system with a diode-array detector (DAD) (Agilent Technologies, Palo Alto, CA, USA); the analytical column was a LiChrospher 100 C18 end-capped (250 × 4.6 mm ID, 5 μm) (Merck KGaA, Darmstadt, Germany) eluted at flow rates of 1.0 mL/min with CH_3_CN 0.1% TFA (solvent A)/H_2_O 0.1% TFA (solvent B) in a gradient mode ratio (20% A to 80% A in 20 min). The column effluent was monitored via DAD at 226, 234 and 254 nm (with 800 nm as the reference wavelength). The purity of the sample was evaluated as the percentage ratio between the areas of the main peak and of possible impurities at all the wavelengths, and also using a DAD purity analysis of the chromatographic peak. The purity of compound was found to be greater than 95%. Compound **5** was synthesized as previously described [22].

Compound **4**. The solution of *tert*-butyldimethyl(2-propynyloxy)silane (4.2 mL, 20.8 mmol) in dry THF (50 mL) under positive N_2_ pressure was cooled at −70 °C, and the solution of BuLi (1.6 M) in hexane (14.5 mL, 23.2 mmol) was added dropwise while maintaining the temperature below −65 °C. After completing the addition, the reaction mixture was stirred at −70 °C for additional 30 min, then the solution of (*tert*-butyldimethylsilyloxy)acetaldehyde (3.3 g, 18.8 mmol) in dry THF (20 mL) was added dropwise. After this addition was completed, the cooling bath was removed and the reaction mixture was allowed to reach room temperature. Then, the saturated NH_4_Cl solution (50 mL) was added dropwise, and the organic phase was separated. The aqueous phase was extracted with EtOAc (2 × 50 mL) and the combined organic extracts were washed with brine, dried and evaporated. The obtained oil was purified via flash column chromatography using petroleum ether/EtOAc as the eluents (95/5, *v*/*v*) to yield **4** as a colorless oil. Yield 5.8 g, 88%. ^1^H-NMR (CDCl_3_) δ: 0.10 (s, 3H, SiC*H*_3_), 0.10 (s, 3H, SiC*H*_3_), 0.12 (m, 6H, Si(C*H*_3_)_2_), 0.91 (s, 9H, *t*-Bu), 0.91 (s, 9H, *t*-Bu), 2.58 (d, *J^3^_HH_* = 4.8 Hz, 1H, O*H*), 3.65 (dd, *J^2^_HH_* = 10.0 Hz, *J^3^_HH_* = 7.2 Hz, 1H, CH(OH)C*H*H), 3.77 (dd, *J^2^_HH_* = 10.0 Hz, *J^3^_HH_* = 3.8 Hz, 1H, CH(OH)CH*H*), 4.35 (m, 2H, C*H*_2_C≡C), 4.43 (m, 1H, CH(OH)); ^13^C-NMR (CDCl_3_) δ: −5.4, −5.2, 18.3, 25.8, 51.7, 63.1, 66.9, 82.5, 84.1. MS (ESI^+^) *m/z* 362.4 (M + NH_4_)^+^.

Compound **6**. To the solution of PPh_3_ (5.80 g, 22.1 mmol) in dry THF (150 mL) placed under positive N_2_ pressure, DIAD (4.60 mL, 22.0 mmol) was added in one portion at −10 °C. The reaction mixture was stirred in ice-salt bath until abundant white precipitate blocked stirring. At this point, ice-salt bath was removed and phenol solution **5** (6.60 g, 14.7 mmol) in dry THF (50 mL) was added dropwise, followed by alcohol **4** (5.0 g, 14.5 mmol) solution in dry THF (30 mL). The reaction mixture was stirred at room temperature for 1 h, then it was heated at 60 °C for additional 4 h. The solvent was evaporated under reduced pressure, and the obtained oily residue was purified via flash column chromatography using petroleum ether / EtOAc as the eluents (98/2, *v*/*v*) to yield **6** as a colorless oil. Yield 7.55 g, 67%. ^1^H-NMR (CDCl_3_) δ: −0.02 (s, 3H, SiC*H*_3_), −0.02 (s, 3H, SiC*H*_3_), 0.06 (s, 3H, SiC*H*_3_), 0.07 (s, 3H, SiC*H*_3_), 0.82 (s, 9H, *t*-Bu), 0.87 (s, 9H, *t*-Bu), 1.00 (s, 9H, *t*-Bu), 1.38 (t, *J^3^_HH_* = 7.2 Hz, 3H, CH_2_C*H*_3_), 2.88 (dt, *J^2^_HH_* = 13.8 Hz, *J^3^_HH_* = 6.9 Hz, 1H, C*H*H), 3.20 (dt, *J^2^_HH_* = 13.8 Hz, *J^3^_HH_* = 6.5 Hz, 1H, CH*H*CH_2_), 3.84 (m, 2H, C*H*_2_), 3.90 (dd, *J^2^_HH_* = 10.3 Hz, *J^3^_HH_* = 5.2 Hz, 1H, C*H*HCH), 3.98 (dd, *J^2^_HH_* = 10.3 Hz, *J^3^_HH_* = 5.9 Hz, 1H, CH*H*CH), 4.15 (dd, *J^2^_HH_* = 15.2 Hz, J^5^_HH_ = 1.4 Hz, 1H, C*H*HC≡CCH), 4.18 (dd, *J^2^_HH_* = 15.2 Hz, *J^5^_HH_* = 1.4 Hz, 1H, CH*H*C≡CCH), 4.43 (m, 2H, C*H*_2_CH_3_), 4.82 (m, 1H, C*H*CH_2_), 7.04 (m, 1H), 7.33 (m, 5H), 7.38 (m, 2H), 7.58 (m, 4H), 7.66 (m, 1H) (2C_6_*H*_5_ + C_6_*H*_3_); ^13^C-NMR (CDCl_3_) δ: −5.4, −5.3, −5.3, 14.3, 18.1, 18.4, 19.1, 25.7, 25.9, 26.8, 33.3, 51.6, 60.9, 63.9, 66.0, 74.8, 80.8, 86.7, 123.6, 125.7, 127.6, 129.5, 129.7, 133.8, 134.9, 135.1, 135.6, 154.9, 166.8. MS (ESI^+^) *m/z* 798.0 (M + Na)^+^.

Compound **8**. To the solution of compound **6** (7.55 g, 9.76 mmol) in THF (50 mL), a 10% solution of HCl (8 mL) was added, and the resulting suspension was stirred vigorously for 4 h (after ~2 h solution became clear). Then, the saturated NaHCO_3_ solution (50 mL) was added and the organic solvent was separated. The aqueous phase was extracted with EtOAc (2 × 50 mL). The combined organic extracts were washed with brine, dried and the solvent was removed under reduced pressure. The obtained oil was dissolved in THF (100 mL) and 0.1 N LiOH solution (100 mL) was added. The reaction mixture was stirred for 3 h, then 1 N HCl solution (12 mL) was added and the resulting mixture was extracted with EtOAc (2 × 150 mL). The organic extracts were washed with brine, dried, and the solvent was removed under reduced pressure. The obtained oil (**7**) was dissolved in Ac_2_O (20 mL) and the mixture was heated at reflux for 1 h. Then, the solvent was removed under reduced pressure and the obtained oil was purified via flash column chromatography using petroleum ether/EtOAc as the eluents (8/2, *v*/*v*) to yield a title compound **8** as a colorless oil. Yield 2.45 g, 46% for three steps. ^1^H-NMR (CDCl_3_) δ: 1.02 (s, 9H, *t*-Bu), 2.05 (s, 3H, COC*H*_3_), 2.83 (dt, *J^2^_HH_* = 13.4 Hz, *J^3^_HH_* = 6.9 Hz, 1H, C*H*HCH_2_O), 3.26 (dt, *J^2^_HH_* = 13.4 Hz, *J^3^_HH_* = 5.9 Hz, 1H, CH*H*CH_2_O), 3.81 (m, 1H, CH_2_C*H*HO), 3.91 (m, 1H, CH_2_CH*HO*), 4.20 (dd, *J^2^_HH_* = 13.4 Hz, *J^3^_HH_* = 10.0 Hz, 1H, C*H*HCH), 4.32 (dd, *J^2^_HH_* = 13.4 Hz, *J^3^_HH_* = 4.5 Hz, 1H, CH*H*CH), 4.59 (m,2H, C*H*_2_C≡C), 5.32 (m, 1H, C*H*CH_2_), 7.19 (m, 1H), 7.38 (m, 6H), 7.50 (m, 3H), 7.61 (m, 3H) (2C_6_*H*_5_ + C_6_*H*_3_); ^13^C-NMR (CDCl_3_) δ: 19.1, 20.5, 26.7, 33.3, 51.6, 63.6, 66.6, 70.4, 80.1, 83.3, 124.4, 124.8, 127.6, 129.6, 129.8, 133.4, 133.6, 135.4, 135.5, 136.9, 149.8, 169.2, 169.9. MS (ESI^+^) *m/z* 565.6 (M + Na)^+^.

Compound **9**. To the solution of **8** (2.0 g, 3.8 mmol) in dry THF (50 mL), pyridine (2.5 mL, 3.5 mmol) was added, followed by Py⋅(HF)_x_ complex (2.0 mL, 77 mmol). The reaction mixture was stirred at room temperature for 6 h, then diluted with EtOAc (100 mL) and washed with the saturated citric acid solution (50 mL). The aqueous phase was extracted with EtOAc (50 mL) and the combined organic extracts were washed with H_2_O (50 mL), saturated NaHCO_3_ solution (2 × 50 mL), brine, dried and the solvent was removed under reduced pressure. The obtained oil was purified via flash column chromatography using petroleum ether/EtOac as the eluents (6/4, *v*/*v*) to yield **9** as a colorless oil. Yield 1.05 g, 90%. ^1^H-NMR (CDCl_3_) δ: 2.11 (s, 3H, COC*H*_3_), 2.94 (dt, *J^2^_HH_* = 13.8 Hz, *J^3^_HH_* = 6.9 Hz, 1H, C*H*HCH_2_OH), 3.21 (dt, *J^2^_HH_* = 13.8 Hz, *J^3^_HH_* = 6.9 Hz, 1H, CH*H*CH_2_OH), 3.88 (m, 2H, C*H*_2_OH), 4.27 (dd, *J^2^_HH_* = 13.8 Hz, *J^3^_HH_* = 10.5 Hz, 1H, C*H*HCH), 4.38 (dd, *J^2^_HH_* = 13.8 Hz, *J^3^_HH_* = 4.5 Hz, 1H, CH*H*CH), 4.67 (dd, *J^2^_HH_* = 15.9 Hz, *J^5^_HH_* = 1.7 Hz, 1H, C*H*HC≡CCH), 4.71 (dd, *J^2^_HH_* = 15.9 Hz, *J^5^_HH_* = 1.7 Hz, 1H, CH*H*C≡CCH), 5.38 (m, 1H, C*H*CH_2_), 7.24 (m, 1H), 7.53 (m, 1H), 7.61 (m, 1H) (C_6_*H*_3_); ^13^C-NMR (CDCl_3_) δ: 20.6, 33.0, 51.7, 62.6, 66.5, 70.6, 80.0, 83.6, 124.7, 125.2, 129.9, 133.1, 135.9, 149.8, 169.0, 170.2. MS (ESI^+^) *m/z* 327.4 (M + Na)^+^.

Compound **10**. To the solution of alcohol **9** (1.0 g, 3.3 mmol) in acetone (80 mL) placed in ice bath, 2.5 M Jones reagent (1.5 mL, 3.75 mmol) was added. An abundant green precipitate is formed. The ice bath was removed and the reaction was stirred at room temperature for 2 h. Then, the reaction mixture was diluted with H_2_O (150 mL) and extracted with EtOAc (1 × 100 mL, 2 × 50 mL). The combined organic extracts were washed with brine, dried and the solvent was removed under reduced pressure. The yellow oil obtained was purified via flash column chromatography using DCM/MeOH as the eluents (95/5, *v*/*v*) to yield **10** as a colorless oil. Yield 0.79 g, 75%. ^1^H-NMR (CDCl_3_) δ: 2.05 (s, 3H, COC*H*_3_), 3.76 (d, *J^2^_HH_* = 16.9 Hz, 1H, C*H*HCOOH), 3.91 (d, *J^2^_HH_* = 16.9 Hz, 1H, CH*H*COOH), 4.24 (dd, *J^2^_HH_* = 13.8 Hz, *J^3^_HH_* = 10.0 Hz, 1H, C*H*HCH), 4.38 (dd, *J^2^_HH_* = 13.8 Hz, *J^3^_HH_* = 4.1 Hz, 1H, CH*H*CH), 4.63 (dd, *J^2^_HH_* = 15.8 Hz, *J^5^_HH_* = 1.7 Hz, 1H, C*H*HC≡CCH), 4.66 (dd, *J^2^_HH_* = 15.8 Hz, *J^5^_HH_* = 1.7 Hz, 1H, CH*H*C≡CCH), 5.35 (m, 1H, C*H*CH_2_), 7.23 (m, 1H), 7.51 (m, 1H), 7.65 (m, 1H) (C_6_*H*_3_); ^13^C-NMR (CDCl_3_) δ: 20.5, 34.8, 51.6, 66.5, 70.8, 79.5, 83.6, 124.2, 125.0, 128.0, 131.0, 136.5, 149.8, 168.7, 170.3, 176.3. MS (ESI^+^) *m/z* 317.3 (M + H)^+^.

Compound **12**. To the solution of phenylacetic acid **10** (400 mg, 1.25 mmol) in dry DCM (20 mL) DMAP (450 mg, 3.70 mmol) and Meldrum’s acid (260 mg, 1.82 mmol) were added followed by EDCI⋅HCl (360 mg, 1.82 mmol). The reaction mixture was stirred at room temperature for 4 h, then it was diluted with DCM and the organic phase was washed with the saturated oxalic acid solution (20 mL), brine, and dried and then the solvent was evaporated under reduced pressure. The obtained brown oil (**11**) was dissolved in a mixture of toluene (10 mL) and MeOH (1 mL) and the reaction was heated under reflux for 2 h. The organic solvent was removed under reduced pressure and the obtained oil was purified via flash column chromatography using petroleum ether/EtOAc (from 7/3 to 6/4, *v*/*v*) to yield a title compound **12** as a colorless oil. Yield 250 mg, 54%. ^1^H-NMR (CDCl_3_) δ: 2.09 (s, 3H, COC*H*_3_), 3.57 (d, *J^2^_HH_* = 15.8 Hz, 1H, C*H*HCO), 3.59 (d, *J^2^_HH_* = 15.8 Hz, 1H, CH*H*CO), 3.74 (s, 3H, OC*H*_3_), 3.87 (d, *J^2^_HH_* = 17.5 Hz, 1H, C*H*HCOO), 3.91 (d, *J^2^_HH_* = 17.5 Hz, 1H, CH*H*COO), 4.27 (dd, *J^2^_HH_* = 13.8 Hz, *J^3^_HH_* = 9.6 Hz, 1H, C*H*HCH), 4.41 (dd, *J^2^_HH_* = 13.8 Hz, *J^3^_HH_* = 4.1 Hz, 1H, CH*H*CH), 4.66 (dd, *J^2^_HH_* = 15.8 Hz, *J^5^_HH_* = 1.4 Hz, 1H, C*H*HC≡C), 4.69 (dd, *J^2^_HH_* = 15.8 Hz, *J^5^_HH_* = 1.7 Hz, 1H, CH*H*C≡C), 5.31 (m, 1H, C*H*CH_2_), 7.25 (m, 1H), 7.44 (m, 1H), 7.67 (m, 1H) (C_6_*H*_3_); ^13^C-NMR (CDCl_3_) δ: 20.6, 43.9, 48.6, 51.6, 52.4, 66.6, 70.8, 79.7, 83.8, 124.2, 125.2, 128.0, 131.2, 136.7, 149.9, 167.4, 168.6, 170.0, 199.6. MS (ESI^+^) *m*/*z* 373.4 (M + H)^+^.

Compound **1**. To the solution of β-ketoester **12** (0.46 g, 1.23 mmol) in dry THF (12 mL) acenaphtenechinone (0.25 g, 1.37 mmol) and MeOH (4 mL) were added, followed by Et_3_N (0.26 mL, 1.87 mmol). The resulting suspension was vigorously mixed at room temperature for 20 h. Then, the solvents were removed and the resulting brown oil was dissolved in Ac_2_O (3 mL). The obtained solution was placed in an ice bath and 5 drops of 98% H_2_SO_4_ were added. The color of the reaction mixture changed from brown-yellow to dark violet. After 10 min at 0 °C, cold MeOH (20 mL) was added and stirring was continued for additional 10 min. The dark precipitate was filtered, washed with a small amount of cold MeOH and dried on filter. Finally, compound **1** was crystallized from hot EtOAc, filtered and dried to yield the title compound as a dark-purple crystalline solid. Yield 450 mg, 70%. ^1^H-NMR (CDCl_3_) δ: 1.98 (s, 3H, COC*H*_3_), 4.00 (s, 3H, OC*H*_3_), 4.13 (m, 1H), 4.31–4.50 (m, 3H) (2C*H*_2_), 5.09 (m, 1H, C*H*CH_2_), 7.47–7.82 (m, 6H), 7.94 (m, 1H), 8.08 (m, 1H), 8.75 (m, 1H) (C_10_*H*_6_ + C_6_*H*_3_); ^13^C-NMR (CDCl_3_) δ: 20.3, 51.5, 51.7, 66.4, 70.4, 78.9, 84.6, 110.0, 117.7, 124.5, 125.7, 125.8, 126.3, 127.7, 128.0, 128.7, 129.8, 130.4, 131.1, 131.2, 131.3, 136.5, 145.7, 149.6, 154.1, 162.8, 168.9, 169.9, 170.5, 196.9. MS (ESI^+^) *m*/*z* 543.2 (M + Na)^+^.

#### 4.1.2. Stability Assay

The evaluation of the stability in phosphate-buffered saline (PBS, 50 mM) and in human serum was carried out as reported in [44], using a HP1200 chromatograph system (Agilent Technologies, Palo Alto, CA, USA) described therein. For the stability study of compound **1**, the analytical column was a ZORBAX Eclipse XDB-C8 (150 × 4.6 mm, 5 µm; Agilent) and the mobile phase consisted of 0.1% aqueous TFA and 0.1% TFA CH_3_CN 40/60 *w*/*w*%, eluted at flow rate = 1.0 mL/min. The injection volume was 20 µL (Rheodyne, Cotati, CA, USA). The column effluent was monitored with UV-VIS detection (234 nm referenced against a 800 nm wavelength), and fluorescence detection (excitation and emission wavelength of 490 and 525 nm, respectively, gain factor = 10). Using this RP-HPLC procedure, we separated and quantified compound **1** from any degradation products during the incubation time. Quantification was performed using a calibration curve obtained with standard solutions of **1**, chromatographed under the same experimental conditions, in a concentration range of 1–100 µM (r^2^ > 0.99). Results are expressed as % of unmodified compound during the incubation time. The pseudo-first-order half-times (t_1/2_) for hydrolysis were calculated with exponential decay equation model via Prism v. 5.0 (GraphPad Software Inc., San Diego, CA, USA).

### 4.2. Biology

#### 4.2.1. Cell Lines

Human NCI-H2228, A549, MDA-MB-231, MDA-MB-468, PANC-1, Capan-1, BEAS-2B, MCF10A and HPDE cell lines were purchased from ATCC (Manassas, VA, USA) and grown in the respective culture media containing 10% *v*/*v* fetal bovine serum, 1% *v*/*v* penicillin-streptomycin, and 1% *v*/*v* glutamine. Cell lines were authenticated via microsatellite analysis with the PowerPlex kit (Promega Corporation, Madison, WI, USA; last authentication: September 2022). *Mycoplasma* spp. contamination was assessed every 3 weeks using RT-PCR, and contaminated cells were discarded.

#### 4.2.2. CORM Intracellular Uptake and CO Release

The 1 × 10^6^ cells were washed with phosphate saline buffer (PBS), detached with Cell Dissociation Reagent (Merck KGaA, Darmstadt, Germany), centrifuged at 12,000× *g* for 2 min, resuspended in 500 µL PBS and sonicated. A 50 µL was used to measure cellular proteins; in the remaining aliquot, the intracellular fluorescence caused by the products of hydrolysis of CORM at excitation λ = 355 nm and emission λ = 460 nm was read using a Synergy HT Multi-Detection Microplate Reader (Bio-Tek Instruments, Winooski, VT, USA). Results were expressed as relative fluorescence units (RFUs)/mg of cellular proteins.

#### 4.2.3. Cell Viability

The 1 × 10^5^ cells were seeded and after 24 h they were incubated as indicated in the Results section. After 72 h, viability was measured via the ATPlite Luminescence Assay System (PerkinElmer, Waltham, MA, USA), according to manufacturer’s instructions, using a Synergy HT Multi-Detection Microplate Reader. Results were expressed as relative luminescence units (RLUs); RLUs of untreated cells were considered as 100% viability. The viability of the other experimental conditions was expressed as the percentage of viable cells versus untreated cells.

#### 4.2.4. ABC Transporters Quantification and Activity

The amount of ABC transporters present on the cell surface was measured on 1 × 10^5^ cells washed in PBS, containing 0.5% bovine serum albumin (BSA) and 2 mM EDTA, centrifuged at 300× *g* for 10 min, then treated with Inside Stain Kit (Miltenyi Biotec., Bergisch Gladbach, Germany), as per manufacturer’s instructions. The cells were then incubated 30 min at room temperature with anti-CD243/ABCB1 antibody (PE-Vio^®^ 770-conjugated, REA495; Miltenyi, Bologna, Italy), anti-MRP1/ABCC1 antibody (PE-conjugated, REA481; Miltenyi) or anti-MRP5/ABCC5 antibody (ab24107; Abcam, Cambridge, UK) followed by the secondary anti-rat IgG H&L antibody (Alexa Fluor^®^ 488-conjugated; Abcam, Cambridge, UK). After an additional washing step with the Inside Perm reagent (Miltenyi, Bologna, Italy), cells were analyzed with a Guava^®^ easyCyte flow cytometer (Millipore, Billerica, MA, USA). The percentage of cells positive for each ABC transporter was calculated with InCyte software v. 3.3 (Millipore, Billerica, MA, USA). To measure the activity of ABC transporters, cell membranes were prepared via differential centrifugation as detailed in [45]. Briefly, 500 µg of membrane proteins were immune-precipitated overnight at 4 °C under non-denaturing conditions using the following antibodies (diluted 1:10): anti-P-glycoprotein/ABCB1 (ab129450), anti-MRP1/ABCC1 (ab260038), and anti-MRP5/ABCC5 (ab180724; all from Abcam, Cambridge, UK), using 25 μL Pure Proteome Beads A/G (Millipore, Billerica, MA, USA). The absorbance of hydrolyzed phosphate from ATP, as an index of the catalytic activity of each ABC transporter, was measured spectrophotometrically as reported [44]. Results were expressed in μmol of hydrolyzed phosphate/min/mg of proteins, based on a titration curve with serial dilutions of NaHPO_4_.

#### 4.2.5. Mitochondrial Extraction, ETC and ATP Measurement

Mitochondrial extracts were prepared as detailed in [23]. The electron efflux from complex I to complex III, taken as an index of mitochondrial respiratory activity, was measured on 50 µg of mitochondrial extracts, as reported in [23]. Results were expressed as nanomoles of reduced cytochrome c/min/mg of mitochondrial proteins. Mitochondrial ATP was quantified with the ATP Bioluminescent Assay Kit (FLAA; Merck KGaA, Darmstadt, Germany), as per manufacturer’s instructions. Results were expressed as nanomoles/mg of mitochondrial proteins.

#### 4.2.6. Mitochondrial Depolarization

The 1 × 10^6^ cells were stained with 2 μM of JC-1 fluorescent probe (Biotium Inc., Hayward, CA, USA) for 30 min at 37 °C, centrifuged at 13,000× *g* for 5 min and resuspended in 0.5 mL PBS. Red fluorescence (λ excitation = 550 nm, λ emission = 600 nm), indicating polarized and undamaged mitochondria, and green fluorescence (λ excitation = 485 nm; λ emission = 535 nm), indicating depolarization and damaged mitochondria, were read using a Synergy HT Multi-Detection Microplate Reader, and expressed as relative fluorescence units (RFUs). Results were expressed as the percentage of green (depolarized)/red (polarized) mitochondria [32]. The opening of the mPTP, a second index of mitochondria depolarization and damage, was quantified in 1 × 10^6^ cells via flow cytometry (Guava EasyCyte equipped with the InCyte software v. 3.3, Millipore, Billerica, MA, USA) using the Mitochondrial Permeability Transition Pore Assay Kit (BioVision, Milpitas, CA, USA), as per manufacturer’s instructions. Autofluorescence of unstained cells was subtracted as blank. Results were expressed as the percentage of fluorescent cells over total cells.

#### 4.2.7. ROS Measurement

The 10 × 10^6^ cells were washed with PBS, detached with the Cell Dissociation Reagent (Merck KGaA, Darmstadt, Germany), centrifuged at 12,000× *g* for 2 min, and resuspended in 500 µL PBS. A 50 µL was sonicated and used to measure cellular proteins. The cell suspensions were incubated for 30 min at 37 °C with 5 µM of 5-(and-6)-chloromethyl-2′,7′-dichlorodihydro-fluorescein diacetate (CM-H2DCFDA) (ThermoFisher, Waltham, MA, USA) or with 5 µM MitoSOX (ThermoFisher, Waltham, MA, USA), to measure total or mitochondrial ROS, respectively, with a Synergy HT Multi-Detection Microplate Reader. RFUs were converted to nanomoles ROS/mg of cellular proteins, based on a titration curve obtained with H_2_O_2_ at serial dilutions.

#### 4.2.8. Lipoperoxidation Assay

Cytosolic and mitochondrial extracts were prepared as detailed in [23] from 10 × 10^6^ cells. A 50 µL aliquot from each fraction was used to quantify the proteins. The extent of oxidative damage was assessed on 200 µg per fraction by measuring the amount of thiobarbituric reactive substances (TBARS) with the Lipid Peroxidation (4-HNE) Assay Kit (Abcam) as per manufacturer’s extraction. Results were expressed as nmoles/mg of cytosolic or mitochondrial protein.

#### 4.2.9. Caspase-9 and Caspase-3 Activation

Caspase-9 and caspase-3 activity were measured in 1 × 10^6^ cells with Caspase 9 Fluorometric Assay Kit (Enzo Life Science, Roma, Italy) and Caspase 3 Fluorescence Assay Kit (Cayman Chemical, Ann Arbor, MI, USA), respectively. Results were expressed as nmol of hydrolyzed substrate of caspase-9 or caspase-3/mg of cellular proteins, according to a previous titration curve.

#### 4.2.10. Statistical Analysis

Data are expressed as means ± SD. The results were analyzed using a one-way analysis of variance (ANOVA) test (Statistical Package for Social Science software, v.19 IBM SPSS Statistics). *p* < 0.05 was considered significant.

## 5. Conclusions

In this work, we demonstrated that a novel non-metal CORM, able to release CO, is toxic towards several human cancer cell lines and their drug-resistant counterparts. Compound **1** was efficiently taken up by drug-resistant cancer cell lines and it was able to restore their sensitivity to chemotherapeutic drugs. The CO release and the consequent increase in mitochondrial oxidative stress can rescue the efficacy of several chemotherapeutic drugs in solid cancers that express ABC transporters and are refractory to clinically used chemotherapy. These results demonstrate the importance of CORMs in situations where conventional chemotherapy fails and thus open the horizons towards new combinatorial strategies to achieve chemosensitization and overcome multidrug resistance.

## Data Availability

Data supporting the findings of this study are available from the corresponding authors, K.C. upon reasonable request.

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
