# Peer review of "Use of Enzymatically Activated Carbon Monoxide Donors for Sensitizing Drug-Resistant Tumor Cells"

_ijms, 2023, doi:10.3390/ijms241411258_

Round 1

Reviewer 1 Report

In this manuscript, the authors conducted synthesis and evaluation of a CO donor for their ability to sensitize cancer cells. This manuscript includes a large amount of data and authors should be applauded for carrying out all the work. The research direction is clearly very important. However, more work is needed to meet the standard publication.

Here are some major points:

1.     First of all, the design of donor 1 uses an ester linkage (between the alkyne and the cyclopentadienone moieties), an internal alkyne as dienophile, and the formation of a 6-membered ring after the Diels-Alder reaction. All three are known factors that slow down the intended reaction to release CO (Acc. Chem. Res. 2018, 51, 1377-85. PMID: 29762011; Chem. Eur. J. 2017, 23, 9838-9845. PMID: 28544290). The half-life is expected to be long for the intended Diels-Alder reaction. In the manuscript, the degradation half-life (Figure 1) was said to be about 90 min when incubated in human serum. It needs to be noted that there are three ester groups in the compound designed. All three are prone to esterase-mediated hydrolysis, which would lead to the disappearance of the starting materials, but not necessarily the intended donor activation, Diels-Alder reaction, and then subsequent CO release. To prove the thesis of the entire manuscript, it is imperative that HPLC, or LC-MS studies be conducted to confirm the identity of the degradation product(s). It is also important to quantitatively determine CO production (e.g. using GC or COP-1 of Chis Chang’s paper) at the same time. Such data would allow authors to determine the half-life of the intended D-A reaction, which would lead to CO release, but just the disappearance of the starting material. Only after such rigorous chemistry work, one can interpret the results from the subsequent biology experiments. One would expect to see multiples products from “CORM 1” upon incubation in serum. Along the same line, it is puzzling to see why build an esterase-sensitive donor molecule with three ester groups, instead of just the ester group critical for donor activation for CO release. The rationale for the design was not clear.

2.     It is important that the NMR and MS spectra be provided in the SI file for reviewers to assess the identity of the compounds synthesized.

3.     It is not clear why Compound 3 is shown to have a new formed ester group and a carboxyl group, apparently resulting from ester hydrolysis. Again, the identity of all products from “CORM 1” needs to be rigorously established through NMR and MS.

4.     Table 1 shows the results of “CORM 1” accumulation. However, it is not clear what was measured: “CORM 1” or degradation product(s). If it is “CORM 1,” then how the stability of “CORM 1” is take into consideration. Further, it will be critical to show some HPLC or LC-MS chromatograms to provide evidence supporting the numbers in the table.

5.     Line 596: The method used to measure intracellular CO concentration needs a reference. It is not clear what was measured in determining intracellular “CORM 1” uptake. If uptake was extrapolated from “CO concentration” data, then CO diffusion needs to be taken into consideration. Again, uptake studies need to actually measure “CORM 1” concentration in order to be valid, not a derivative number.

6.     The biology experiments lack a proper control. At least the product after CO release from “CORM 1” needs to be used as negative control in order to eliminate the CO-independent effects from “CORM1.”

7.     Given the current confusing state of studying CO biology using a CORM due to the many CO-independent effects and chemical reactivity of the commonly used CORMs, it is imperative that future studies are based on solid molecular science. This manuscript needs to much more work along this line.

8.     I hope all these do not discourage the authors from conducting follow up work. What is being described is an important direction. The CO field certainly needs more chemists to conduct rigorous molecular science.

Minor points:

1.     The authors of the original paper publishing organic CO donors (Wang et al) did not use the term CORM because of all the problems identified with CORMs. It is suggested that the authors of this paper stay away from the term CORM as well. However, this is merely a suggestion and it is up to the authors to decide what to use except the term CORM-1 has already been used by Motterlini in previous publications.

2.     It is suggested that authors look at the issue of significant numbers in the data listed in Table 1. It is also important to include experimental errors.

3. Since only cell culture work was conducted, it is suggested that the authors refrain from using the term “tumor.” Instead, “cancer cells” are more appropriate.

Reviewer 2 Report

The manuscript titled “Use of enzymatically activated carbon monoxide donors for the 2 sensitizing of drug-resistant tumors” reported a novel non-meta CO-releasing molecule (1), which able to release intracellularly CO and fluorescent degradation products under the action of esterase. It’s first demonstration that CO donors can rescue the efficacy of several chemotherapeutic drugs in solid cancers that express ABC transporters and are refractory to clinically used chemotherapy, which is a very significant study. However, there are some problems with this manuscript, as follows:

1.      Compound 1 releases CO upon esterase enzymes activation and the authors should have designed experiments to monitor CO production so that direct results would better indicate that compound 1 is capable of releasing CO.

2.      In the manuscript, the authors use compound 1 as the experimental group, I think an additional group should be added, using the compound as a control group. The control illustrates that sensitization of different drug-resistant tumor cells works through esterase activation to produce CO rather than the compound molecule itself.

3.      Could all esterase enzymes activate compound 1? And if so, how can the application ensure that compound 1 reaches the designated tumour site before releasing CO, and is it possible that it has already been activated by esterase enzymes during transport?

4.      In the manuscript, the authors studied the stability of compound 1, which was progressively enzymatically cleaved in human serum protein containing esterase, and the half-life of 1 in human serum was found to be nearly two hours. If further consideration is given to testing in live animals, will the residence time of compound 1 in vivo be longer than its half-life in plasma?

5.      In experiments on the effects of compound 1 and chemotherapeutic agents on mitochondrial and total ROS and lipid peroxidation in resistance cell lines, there was no significant mitochondrial ROS/mitochondrial TBARS and total ROS/total TBARS in the Compound alone compared to the Combo group, ask whether CO-mediated mitochondrial oxidative stress can only be produced when the combination is administered, whereas CO alone does not cause mitochondrial oxidative stress?

Round 2

Reviewer 2 Report

With all the improvements the authors have made, the manuscript could now be accepted for the publication.